# MOMENTUM MEETS VIRALITY: A NOVEL METRIC FOR UNMASKING SOCIAL BIAS IN VIRAL TWEETS

## ABSTRACT

*Warning: This paper contains examples that may be offensive.*

Predicting which social media posts will go viral is a critical but complex task in the field of computational social science. Previous studies have utilized various measures to forecast the virality of tweets or Facebook posts, but these approaches exhibit limitations, particularly in the absence of a virality metric that specifically considers social biases. In this paper, we test existing metrics and introduce a new metric, *ViralTweet Score (VTS)*, inspired by principles of momentum from physics to better predict a tweet's virality given that it consists of social biases. We compare this new metric with others, highlighting the advantages and disadvantages of each of them as a virality measurement metric. We release the *ViralTweets Dataset* with **88.8k** Hindi tweets and corresponding virality labels based on our VTS metric. We also show how social biases in posts can influence their potential to go viral. We test our hypothesis that VTS is a better metric using two methodologies and we show how VTS achieves an F1 score of **0.87** based on pairwise evaluation methodology and an overall F1 score of **0.58** based on our clustering-based verification methodology. Our work offers a novel metric for understanding tweet virality for biased tweets and opens the door for more equitable and effective social media analytics by considering the role of social biases in virality.[1]

## 1 INTRODUCTION

In recent years, social media platforms have emerged as powerful tools for information dissemination, opinion formation, and social interaction. With user base beyond 500 million[2], Twitter stands as a prominent arena for public discourse, facilitating real-time communication and interaction on a global scale Malik et al. (2019) (Note: Even though Twitter has undergone a rebranding to become $\mathbb{X}$, the dataset employed in this study was collected from the platform during the period when it retained the name Twitter. Thus, we consistently use the term "Twitter" throughout this paper). The phenomenon of tweet virality Jenders et al. (2013), wherein certain tweets gain widespread attention and engagement within the online community, has garnered significant interest among researchers, practitioners, and policymakers Han (2020).

Furthermore, our research extends beyond mere predictions of tweet virality to examine the potential relationship between tweet virality and the social biases they may harbor. Social bias, which includes both prejudice and stereotypes, can be inadvertently propagated through social media content Ghosh et al. (2021). By analyzing viral tweets and introducing a metric (VTS) that accounts for social bias in virality predictions, we aim to uncover insights into how these biases could influence both tweet virality and broader public discourse.

Although some research has been conducted on the spread of tweets in various contexts, a notable gap exists in understanding its dynamics within Indian social media. India, with its diverse linguistic, cultural, political, and socio-economic landscape, presents a compelling setting for investigating the virality of tweets and their implications for societal implications.

---

[1]All the scripts utilized, datasets, human annotations created in this study will be made publicly available upon acceptance.

[2]https://backlinko.com/twitter-users

Tweets pertaining to topics such as politics, religion, caste, and gender are inherently imbued with socio-cultural biases that may influence their virality and reception among different segments of the users. Exploring the relationship between tweet virality and corresponding social bias categories can shed light on the role of online discourse in perpetuating existing social norms and stereotypes.

In essence, we address two **research questions (RQ)**:

- **RQ1**: '*What specific metric most effectively captures the virality of a tweet compared to others?*'
  - We propose a new *ViralTweet Score* (*VTS*) metric by capturing multiple aspects of a tweet and their evolution over time. We then compare this metric with other existing metrics for virality measurement to compare their effectiveness.

- **RQ2**: '*Do biased tweets exhibit greater virality compared to unbiased tweets, and which metric best captures this phenomenon?*'
  - By analyzing biased tweets through different virality measurement metrics, we explore the relationship between biased tweets and their virality under the umbrella of Hindi tweets and the Indian context. Also, we examine which virality metric better correlates with the bias in the tweets.

Though our study and analysis focus on measuring virality in social media posts for Indian culture and Hindi Language, our approach can be extended to any different social context and language. Specifically, in Indian context, our framework contributes to Indic ecosystem which is relatively under-resourced compared to more extensively studied languages. Despite such challenges offered in low-resource settings, the dataset utilized in this study offers valuable insights into social media dynamics across diverse linguistic and cultural backgrounds. This not only addresses our research questions but hold potential to benefit the society by enhancing understanding of virality in social media posts.

**Our contributions are**:

1. *ViralTweets Dataset*- a dataset containing $88.8k$ Hindi tweets from Indian user accounts and the time series information for various engagements for each tweet in the dataset. The tweets collected are for the period between January-October 2019. The dataset will be publicly released for the benefit of the research community and reproducibility (Section 4).
2. For each instance in *ViralTweets Dataset*, we also release corresponding binary social bias labels along with the possible categories such as *gender, religion, racial, age, disability, socioeconomic, caste, regional and political* corresponding to bias types in Indian cultural context. These labels are collected automatically using majority voting among predictions from different LLMs (Section 4). Also, we release a subset of 3k tweets with human annotation by three annotators for binary bias label, corresponding bias category, tweet topic, toxicity label, and bias rationale.
3. A novel momentum-inspired *ViralTweet Score (VTS)* metric based on the *momentum* of the spread of tweets on the Twitter platform. We demonstrate that this metric offers a 7.89% improvement in accurately classifying more viral tweets compared to existing metrics for predicting tweet virality (Section 5) .
4. Analysis for the virality of tweets based on binary clusters for social bias label showing that the tweets with higher VTS scores get classified into bias cluster more often than tweets with low VTS scores (Section 7.2).

## 2 RELEVANCE TO SOCIETY

Understanding and predicting the virality of tweets is crucial, particularly considering the negative consequences that can arise from tweets spreading rapidly across social media platforms. Viral tweets have the potential to amplify harmful messages, perpetuate misinformation, and fuel online harassment and cyberbullying Amon et al. (2020); Hasan et al. (2021). Individuals who find themselves at the center of attention within a short span of time after their social media posts get viral (often due to controversial or biased tweets) may face severe personal and professional reper-

cussions, including reputation damage, job loss[3], and mental health consequences[4]. Moreover, the spread of viral misinformation Guo et al. (2022); Elmas (2023) through tweets can undermine public trust in information sources and exacerbate societal divisions. By studying the factors that contribute to the virality of tweets, especially those with adverse effects, researchers can develop strategies to mitigate the harmful impacts of viral content and promote healthier and more responsible online discourse.

## 3 RELATED WORKS

This section reviews the existing literature on virality in social media, focusing on different aspects of content spread and virality metrics. We categorize the literature into three main areas: social network dynamics, content virality, and virality metrics.

### 3.1 SOCIAL NETWORKS DYNAMICS

The nature of Twitter as both a social network and a news media platform has significant implications for information spread. Early studies of Twitter have shown that over 85% of trending topics on Twitter relate to headline or persistent news Java et al. (2007); Kwak et al. (2010). This dual nature drives virality mechanisms. Pan et al. (2019) highlight the role of social network homophily in enhancing user occupation predictions through network-based features. Duan et al. (2012) assess how social influence and content quality affect Twitter topic summarization, emphasizing user interactions' importance. Rahimi et al. (2015) explore user geolocation by leveraging text and network context, illustrating how social ties assist in profiling. Together, these studies underscore the complex interplay between social relations and content properties in shaping social media information dissemination.

### 3.2 CONTENT VIRALITY

Content virality has been extensively studied, particularly in the context of images and news. Key studies have proposed various metrics and models to understand, predict content virality:

- **Image Virality:** Works by Deza & Parikh (2015); Dubey & Agarwal (2017) and Guerini et al. (2013) focus on understanding the virality of images. These studies explore the visual characteristics that contribute to virality and propose metrics such as average score or resubmissions to quantify it.
- **News Virality:** The prediction of news virality has been tackled by Lu & Szymanski (2018); Benson (2020), who use various machine learning approaches to predict the spread of news articles based on community structures and textual content, respectively.
- **Social Media Content:** The diffusion of content such as memes and videos has been analyzed in studies like Ling et al. (2021; 2022), which dissect the elements that make such content go viral on platforms like TikTok.

### 3.3 VIRALITY METRICS

The definition and quantification of virality are crucial for both theoretical and practical applications:

- **General Metrics:** Studies such as Kwak et al. (2010) have examined the role of influencers and the importance of metrics like retweets and PageRank in determining the spread of information.
- **Multimodal Metrics:** Research by Wang et al. (2018) and Wong et al. (2023) has developed deeper insights into the virality of content by considering multimodal aspects, integrating both textual and visual data.
- **Emotional and Psychological Aspects:** The impact of emotions on virality is explored in Pröllochs et al. (2021), which links the emotional content of online rumors to their spread and influence.

---

[3]https://www.teenvogue.com/story/intern-fired-racist-n-word-tweet
[4]https://www.nytimes.com/2015/02/15/magazine/how-one-stupid-tweet-ruined-justine-saccos-life.html

The body of work on social media virality is vast and varied, touching on different aspects of social networks, content types, and metrics. This research builds on these foundational studies by proposing a new metric for virality prediction and focusing on the specific context of tweet virality within the Indian social media landscape, aiming to investigate unique characteristics and influences such as social biases.

## 4 VIRALTWEETS DATASET

Data for the ViralTweets Dataset was collected systematically from Twitter using the official Twitter API during the period from January 2019 to October 2019. Due to recent changes to the Twitter API, we were limited in the time period of the tweets we had access to; however, this data, collected in 2020, still provides a robust foundation for exploring social media dynamics. Focusing on Hindi language tweets specifically allows for an in-depth analysis of social media interactions within the Indian context. In total, approximately 9.24 million tweets were initially collected, offering a comprehensive dataset for our research to begin with.

### 4.1 DATA FILTERING AND CLEANING

To ensure the highest reliability and utility of the data included in the ViralTweets Dataset, we implemented a rigorous and meticulous data filtering process. The steps in the process are outlined below, along with their impact on the dataset size:

- Initially, the dataset was refined by removing non-Hindi tweets and duplicates, ensuring that the dataset consisted only of original content. This step reduced the dataset size from 9.24 million to 7.14 million tweets.

- To facilitate a comprehensive time-series analysis, capturing dynamic engagement metrics—*likes, shares, and retweets over time*—we retained only those tweets that had time-series data spanning more than one day. This reduced the dataset further to 200,000 tweets.

- Finally, to ensure the dataset provided adequate time-series information for a meaningful analysis, we implemented an additional filter to include only tweets with at least four distinct time-series data points. This stringent criterion streamlined the dataset to 88,800 tweets, guaranteeing a robust representation of engagement metrics that are critical for model training and detailed analyses of bias and virality.

Figure 2 illustrates the data, showing the retweet count for all tweets in the ViralTweets Dataset. The script for our data filtering pipeline will be made publicly available to ensure reproducibility.

### 4.2 BIAS LABELS

In the domain of social media, bias is pervasive and can subtly influence the dissemination and perception of information. To address this, our study meticulously labels social biases in tweets, utilizing the recent open-source multilingual large language models (LLMs), and a similar dataset released by Sahoo et al. (2023). To ensure the reliability and accuracy of annotations of social biases in tweets, we employed a model voting system involving four multilingual language models: Llama-3.1 (Dubey et al., 2024), Llama-3.1-Instruct[5], and Openhathi-7b[6], and a XLM-Roberta (Conneau et al., 2020) . Each model is trained on diverse datasets and has been fine-tuned for specific capabilities in language understanding and instruction following. Here, we detail the process and rationale for using multiple models in determining the presence of social biases.

**Finetuning Using Sahoo et al. (2023).** The dataset released by Sahoo et al. (2023) has social bias labels (Yes/No) for Hindi tweets and corresponding bias categories, among a few other labels. We fine-tune Llama-3.1, Openhathi, and XLMR[7] models using the binary social bias labels of this dataset. We use the train, dev, and test split provided by the authors of this dataset. The F1-scores

---

[5]https://huggingface.co/meta-llama/Llama-3.1-8B-Instruct

[6]https://huggingface.co/sarvamai/OpenHathi-7B-Hi-v0.1-Base

[7]We use XLMR because the paper has reported that XLMR outperforms other multilingual models.

of these three models for bias label, computed using a test set of Sahoo et al. (2023), are 88.7, 81.2, and 83.4, respectively. We also evaluated these model using the annotated subset (Section 4.3) of our dataset. The F1-scores of Llama, Openhathi, and XLMR on this subset are 86.2, 82.0, and 83.1, respectively. More technical details of fine-tuning are presented in Appendix B.

**Instruction Tuning.** We also instruction tune the Llama-3.1-Instruct model with in-context examples for binary bias prediction. The optimal prompt selection was done using the validation set of the dataset by Sahoo et al. (2023). The F1-score on the test set of Sahoo et al. (2023) and the subset of our dataset are 77.2 and 78.3, respectively. The exact instruction used is provided in Appendix C.

**Voting Procedure.** Each tweet in the ViralTweets Dataset is assigned a bias label by all four models. For a tweet to be labeled with a specific bias label (Yes/No), at least three out of the four models had to agree on the classification. This majority voting approach reduces the likelihood of misclassification due to model-specific biases or errors and ensures a more balanced and accurate assessment.

### 4.3 HUMAN ANNOTATION FOR BIAS

Motivated by the decent performance of LLMs as annotators on multiple languages Pavlovic & Poesio (2024), and due to the large dataset size, we conducted the social bias annotations with the help of three LLMs as described in Section 4.2. However, to assess the quality of the model annotations, we performed human annotations on a small subset of the dataset. We randomly chose 3k tweets from our dataset (say, 3k-subset) and annotated them to check the presence of social bias in them[8].

We present the agreement between each model prediction with the annotations by each of the annotators for 3k-subset in Table 1 along with the inter-annotator agreement between the annotators. The Krippendorff's alpha Krstovski et al. (2022) among the three annotators is 63.3, which is a very good score considering the subjectivity of bias label. Annotator 3 exhibits the highest agreement with each of the model predictions for the bias label, with the highest agreement (Cohen's kappa) of 63.1 with the XLMR model.

Table 1: Agreement between human annotators and the machine prediction. Each value, here, represents the Cohen's kappa score. A1, A2, and A3 represent three annotators. The highest cohen's kappa value, 95.4, is between A2 and A3.

| Models ($\downarrow$) | A1 | A2 | A3 |
|---|---|---|---|
| Llama-3.1 | 61.3 | 60.4 | 62.3 |
| Llama-3.1-Instruct | 50.4 | 56.3 | 57.8 |
| Openhathi | 55.2 | 53.9 | 56.2 |
| XLMR | 57.2 | 57.4 | 63.1 |
| A1 | – | 72.1 | 77.1 |
| A2 | 72.1 | – | 95.4 |
| A3 | 77.1 | 95.4 | – |

### 4.4 DATASET CHARACTERISTICS

The final dataset comprises 88.8k unique Hindi tweets. These tweets are distributed across various categories of social biases, as shown in Table 5 of Appendix A. We will release the *ViralTweets Dataset* with binary bias label predictions, bias categories from different models, and different VTS scores for each tweet. Also, we will release the 3k-subset dataset with human annotations for binary bias label, possible bias categories (gender, religion, racial, age, disability, socioeconomic, caste, regional, and political), sentiment of the tweet (positive, negative, neutral), relevant topic of the tweet (politics, sports, entertainment, violence, religion, and others), toxicity label for each tweet (toxic, offensive, misogyny, hate speech, and neutral), and the rationale behind the bias label. Annotators are asked to write a free text describing the reason behind the bias label, if annotated for the presence of any social bias, as the rationale. In future, this extensive dataset can be used by the researchers for more nuanced analysis of bias in social media.

---

[8]More details on the annotation are discussed in Appendix D.

### 4.4.1 ENGAGEMENT METRICS.

Each tweet in the dataset is associated with detailed engagement metrics including likes, retweets, replies, and the time series of these interactions.

## 4.5 DATA USAGE

The ViralTweets Dataset is designed for research into how tweets become popular, especially in India. It helps researchers study how biases in society show up in popular tweets and what makes tweets go viral. The dataset also gives detailed information on how people interact with these tweets.

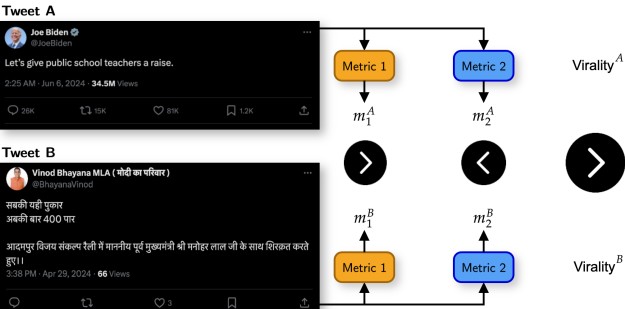

Figure 1: Hypothesis Verification, to compare ViralTweet score with other existing metrics. We train two models to predict two different virality metrics. Each model once trained, scores two tweets for their respective metric. The order relation of the two tweets are used as a prediction to compare with ground truth.

## 5 VIRALTWEET SCORE

The ViralTweet Score Metric is introduced to quantify the virality of tweets based on their engagement dynamics over time. Unlike traditional metrics that may consider static counts of likes or retweets, this metric emphasizes the rate of change in engagement, capturing the momentum of a tweet's spread. This approach is particularly useful for understanding how quickly a tweet gains popularity, which is a critical aspect of virality in fast-paced social media environments.

## 5.1 DEFINITION

Given a tweet $Tweet[i]$ in the dataset, we have time-series data for $T + 1$ timestamps detailing various engagement metrics and user account information. Each metric offers insight into how the tweet is being received and shared among users:

- **likes** (*favourite_count*): Total number of times the tweet has been liked by users.
- **retweet_count**: Total number of times the tweet has been retweeted without modifications.
- **follower_count**: Total number of followers of the tweet creator on the platform. This is the only metric used from user account information.

## 5.2 METRIC FORMULATION

The *ViralTweet Score* (VTS) Metric is formulated to consider the growth in engagement metrics over successive timestamps. Specifically, we focus on the change in 'likes' ($L$) and 'retweets' ($R$), as these are strong indicators of a tweet's reach and endorsement. As the metric is inspired by the momentum concept from physics, it has a *mass* component and a *velocity* component. The *velocity* component is measure as the change in its value over time. The different variations of VTS metric for any given time $t + 1$ are defined as:

$$(\mathbf{VTS}_{F \times (v_L + v_R)})_{t+1} = followers_{t+1} \times \left[ \frac{likes_{t+1} - likes_t}{time_{t+1} - time_t} + \frac{retweets_{t+1} - retweets_t}{time_{t+1} - time_t} \right] \tag{1}$$

$$(\mathbf{VTS}_{F \times (0.6\, v_L + 0.4\, v_R)})_{t+1} =$$
$$followers_{t+1} \times \left[ \alpha * \frac{likes_{t+1} - likes_t}{time_{t+1} - time_t} + (1 - \alpha) * \frac{retweets_{t+1} - retweets_t}{time_{t+1} - time_t} \right] \tag{2}$$

$$(\mathbf{VTS}_{0.6\, L \times 0.4\, v_R})_{t+1} = \alpha * likes_{t+1} \times (1 - \alpha) * \frac{retweets_{t+1} - retweets_t}{time_{t+1} - time_t} \tag{3}$$

Where:

- $likes_{t+1}$ is the number of likes at time $t + 1$.
- $retweet_{t+1}$ and $retweet_t$ are the retweet counts at times $t + 1$ and $t$, respectively.
- $time_{t+1}$ and $time_t$ represent the corresponding timestamps.
- $\alpha$ is a learnable parameter. However, we set $\alpha$ to be $0.6$[9].

Other variations of VTS scores we experimented with are presented in the Appendix E.

This calculation captures the velocity of engagement—a key aspect of virality. By multiplying the likes by the rate of change in retweets, we obtain a measure that reflects both the popularity and the propagation speed of the tweet.

### 5.3 Overall ViralTweet Score

To assess the overall *VTS* of a tweet over the available data period, we average the individual ViralTweet score calculations across all timestamps:

$$VTS = \sum_{t=1}^{T} \frac{ViralTweetScore_{t+1}}{T} \tag{4}$$

Here $T$ is the number of points from the training dataset, used for calculating ViralTweet Score. This aggregated measure provides a single score that can be used to compare the virality of different tweets within our dataset.

## 6 Methodology

In this section, we outline the methodologies employed to rigorously evaluate and compare the effectiveness of VTS with other metrics in predicting tweet virality. Our aim is to determine which metric serves as a more reliable and accurate indicator to predict virality of potential biased content on social media platforms.

We test our hypothesis that VTS is a superior metric using two distinct methodologies:

- First, we examine whether a model trained to predict VTS can accurately discern which of two posts is likely to be more viral. (Section 6.1)
- Second, we test in the unsupervised setting, whether VTS provides more accurate predictions of virality and bias, when compared to other virality metrics for newly encountered tweets. (Section 6.2)

---

[9]This is because the internal algorithm of twitter (https://tinyurl.com/54u4mrc8) gives relatively 0.6 weightage to like count and 0.4 to retweet count.

Table 2: Comparative performance analysis of XLM-Roberta, mT0-large, and Sarvam-2b[10] across various virality metrics. This table showcases the mean squared error (MSE), root mean squared error (RMSE), R-squared (R2), and mean absolute error (MAE) for each model and metric. **Bold-face** values indicate the best metric among the compared ones. The upward arrow ↑ indicates that a higher value corresponds to a better metric; the downward arrow ↓ indicates that a lower value corresponds to a better metric.

| Metric | XLM-Roberta | | | | mT0-large | | | | sarvam-2b | | | |
|---|---|---|---|---|---|---|---|---|---|---|---|---|
| | MSE ↓ | RMSE ↓ | R2 ↑ | MAE ↓ | MSE ↓ | RMSE ↓ | R2 ↑ | MAE ↓ | MSE ↓ | RMSE ↓ | R2 ↑ | MAE ↓ |
| $VTS_{F\times(v_L+v_R)}$ | 0.38 | 0.61 | 0.61 | 0.46 | 0.42 | **0.64** | **0.58** | 0.49 | 0.32 | **0.57** | **0.67** | 0.43 |
| $VTS_{F\times(0.6\,v_L+0.4\,v_R)}$ | **0.36** | **0.60** | **0.63** | **0.45** | **0.41** | **0.64** | **0.58** | **0.49** | **0.32** | **0.57** | **0.67** | **0.42** |
| $VTS_{0.6\,L\times0.4\,v_R}$ | 0.52 | 0.72 | 0.47 | 0.55 | 0.55 | 0.74 | 0.43 | 0.57 | 0.51 | 0.71 | 0.48 | 0.54 |
| Likes | 0.52 | 0.72 | 0.47 | 0.54 | 0.55 | 0.74 | 0.44 | 0.56 | 0.48 | 0.69 | 0.51 | 0.53 |
| Retweets | 0.69 | 0.83 | 0.30 | 0.64 | 0.72 | 0.85 | 0.27 | 0.66 | 0.68 | 0.82 | 0.31 | 0.64 |
| Retweets / Followers | 0.53 | 0.73 | 0.46 | 0.55 | 0.48 | 0.69 | 0.52 | 0.53 | 0.36 | 0.60 | 0.64 | 0.45 |

## 6.1 PAIR-WISE COMPARISON OF TWEETS

The first analytical method to test performance of virality metrics in this study is the pair-wise comparison of tweets. This approach allows us to directly compare the ViralTweet Score of two tweets and determine which one exhibits greater virality under similar conditions. Each pair of tweets is selected based on having comparable initial conditions such as similar posting times, initial user engagement, or demographic reach. Figure 1 gives an overview of how we compare two models finetuned to predict two different metrics using the test data, to assess the "goodness" of a metric for virality prediction.

### 6.1.1 SELECTION CRITERIA

Tweets are paired using the following criteria to ensure fairness and relevance in comparisons:

- **Temporal Proximity**: Tweets posted within similar time frames are compared to control for variations in user online activity.
- **Initial Engagement**: Tweets with similar initial engagement metrics (likes, retweets within the first hour of posting) but an order of magnitude difference, after one day period are paired to normalize starting popularity.

Using the ViralTweet Score defined earlier, we predict the future virality of tweets based on early engagement data. This involves calculating the ViralTweet Score for each tweet at successive time intervals and using machine learning models to predict its trajectory.

### 6.1.2 MODELS USED.

A set of predictive models, including time series analyses and regression algorithms, are employed to forecast the ViralTweet Score based on initial engagement metrics. These models are trained on historical data from the ViralTweets Dataset, learning patterns of virality that are not immediately apparent to human observers.

Table 3: Comparison of metrics as a predictor of virality for models XLM-Roberta, mT0-large trained to predict different virality metrics. Here P, R, and F1 are Precision, Recall, and F1 scores, respectively.

| Metric | XLM-Roberta | | | mT0-large | | |
|---|---|---|---|---|---|---|
| | P ↑ | R ↑ | F1 ↑ | P ↑ | R ↑ | F1 ↑ |
| $VTS_{F\times(0.6\,v_L+0.4\,v_R)}$ | **0.60** | **0.68** | **0.63** | **0.82** | **0.87** | **0.84** |
| Likes | 0.55 | 0.62 | 0.58 | 0.68 | 0.73 | 0.70 |
| Retweets | 0.58 | 0.65 | 0.61 | 0.76 | 0.79 | 0.78 |
| Retweet / Follower | 0.50 | 0.57 | 0.53 | 0.65 | 0.71 | 0.67 |

## 6.2 Unsupervised prediction of Virality

The second analytical method to test the performance of virality metrics in this study is the unsupervised prediction of bias and virality as a classification task.

For this analysis, we divided our data into training and testing sets. We employed different multilingual models (detailed in Section 7) to encode the tweets. All nine bias categories were consolidated into two categories: 'biased' for any tweet falling into one of the bias categories and 'no-bias' for tweets without identified biases. Similarly, we classified the top $K\%$ ($K$ can be 10, 15, 20, 25 as shown in Table 4) of tweets as 'viral' based on each virality metric, with the rest categorized as 'non-viral'. We then utilized clustering techniques on the tweet embeddings to identify cluster centers. For the test data, tweets were assigned to one of these four clusters based on their closest similarity to the cluster centers. The motivation behind this approach is to explore whether unsupervised methods, which do not rely on predefined labels, can effectively distinguish between viral and non-viral content and how they behave for biased and unbiased tweets using different metrics. This methodology allows us to evaluate the robustness and generalizability of each metric across various contexts without the potential biases introduced by supervised learning labels.

## 7 Experiments, Results and Analysis

In this section, we describe how we evaluate our proposed metric ViralTweet Score VTS and show the precision, recall and F1 scores for predicting virality. We also show how virality correlates with social biases and report precision, recall and F1 scores for classifying virality scores into bias clusters.

We finetune different models such as XLM-Roberta Conneau et al. (2020), mT0 Muennighoff et al. (2023), and sarvam-2b[11] to predict a given metric. We then, compare two models of the same type, trained to predict two different metrics to assess the better predictor of virality from among the two.

Figure 1 gives an overview of how we compare two models finetuned to predict two different metrics using the test data, to assess the "goodness" of a metric for virality prediction. As detailed in Sec 6, using this method, we can now compare two different models trained to predict two different metrics to compare the metrics and the accuracy, precision, and F1 score for the correct order prediction among the two metrics.

We also establish the correlation between virality predictions and social biases for tweets. The motivation to do so is that many of the tweets in this data are not neutral, and the conversations span a limited set of topics, including politics, news, media, and opinionated statements.

### 7.1 Variations of VTS Score and baselines

In experiments, we conduct evaluations with three major variants of VTS Score: $\mathbf{VTS}_{F \times (v_L + v_R)}$: VTS with followers and velocity of likes and retweets, $\mathbf{VTS}_{F \times (0.6\,v_L + 0.4\,v_R)}$: VTS with followers and weighted velocity of likes and retweets, $\mathbf{VTS}_{0.6\,L \times 0.4\,v_R}$: VTS with likes and weighted velocity of retweets. As baseline approaches, we consider metrics like average likes, average retweets, and average retweets/follower count and report precision, recall and F1 of all the metrics (see Table 4) for details.

Formulation and results corresponding to other variants such as $\mathbf{VTS}_{L \times v_R}$: VTS with likes and velocity of retweets, $\mathbf{VTS}_{R \times v_L}$: VTS with retweets and velocity of likes, $\mathbf{VTS}_{0.4\,R \times 0.6\,v_L}$: VTS with retweets and weighted velocity of likes, and average likes/follower count are presented in Appendix.

### 7.2 Results and Analysis

**Evaluating effectiveness of ViralTweet Score for detecting virality.** We evaluate the effectiveness of different virality metrics, emphasizing the performance of the ViralTweet Score (VTS) against traditional metrics such as Likes and Retweets. We explore how these metrics handle the dynamics of social media engagement and their interaction with social biases.

---

[11]We used sarvam model as it is pretrained majorly on Hindi corpus.

Table 4: Performance metrics (Precision, Recall, F1 Score) for different virality metrics across various Top $K\%$ thresholds. The hypothesis is that the top $K\%$ tweets based on each metric value are considered viral.

| Top $K\% \rightarrow$ | | 10% | | | 15% | | | 20% | | | 25% | |
|---|---|---|---|---|---|---|---|---|---|---|---|---|
| Metric | P | R | F1 | P | R | F1 | P | R | F1 | P | R | F1 |
| $\mathbf{VTS}_{F \times (v_L + v_R)}$ | 0.61 | 0.54 | 0.56 | 0.64 | 0.55 | 0.58 | 0.61 | 0.55 | 0.57 | 0.60 | 0.55 | 0.56 |
| $\mathbf{VTS}_{F \times (0.6\,v_L + 0.4\,v_R)}$ | 0.61 | 0.56 | 0.57 | 0.63 | 0.55 | 0.58 | 0.61 | 0.55 | 0.57 | 0.59 | 0.55 | 0.56 |
| $\mathbf{VTS}_{0.6\,L \times 0.4\,v_R}$ | 0.61 | 0.50 | 0.53 | 0.65 | 0.49 | 0.54 | 0.62 | 0.50 | 0.53 | 0.56 | 0.47 | 0.49 |
| **Likes** | 0.63 | 0.49 | 0.53 | 0.66 | 0.49 | 0.54 | 0.62 | 0.50 | 0.53 | 0.59 | 0.49 | 0.52 |
| **Retweets** | 0.61 | 0.47 | 0.51 | 0.65 | 0.47 | 0.52 | 0.61 | 0.47 | 0.51 | 0.57 | 0.47 | 0.50 |
| **Retweets / Follower** | 0.63 | 0.54 | 0.55 | 0.66 | 0.54 | 0.56 | 0.62 | 0.53 | 0.55 | 0.60 | 0.53 | 0.54 |

Our findings show that VTS, which captures the rate of engagement growth, consistently outperforms traditional metrics. This is evidenced by lower mean squared error (MSE), root mean squared error (RMSE), and higher R-squared (R2) values, highlighting its ability to capture the quick shifts in social media engagement. Traditional metrics, while popular, fail to account for the temporal aspects crucial for understanding virality, as seen in Table 2. Also, Table 3 shows the effectiveness of *VTS* based on pair-wise comparison of tweets.

Additionally, precision, recall, and F1 scores from Table 2 support the superiority of VTS. This metric not only minimizes error but also excels in scenarios that require comparative analysis of virality, proving to be a more reliable indicator for researchers and practitioners.

**Classification of VTS scores into bias clusters.** `VTS` shows high precision in detecting non-biased non-viral tweets and good recall in identifying non-biased viral tweets, suggesting its effectiveness in recognizing potential virality without the influence of bias.

Finally, Table 4 confirms the robust performance of VTS and Retweet/Follower count across multiple measures, including accuracy, precision, recall, and F1 score. This comprehensive performance underlines the utility of dynamic metrics like VTS, especially in analyzing the effects of social biases on virality.

These insights affirm that VTS effectively addresses our research questions, proving to be the most effective metric in capturing tweet virality and its interaction with bias, and providing valuable insights for social media analytics.

# 8 CONCLUSIONS AND FUTURE WORK

This study rigorously evaluated various virality metrics, with a special focus on the ViralTweet Score (VTS), across multiple advanced NLP models. Our findings demonstrate that VTS is superior in predicting tweet virality, surpassing traditional metrics like Likes, Retweets, and Retweets per Follower count. This metric's capacity to capture the dynamic changes in social media engagement makes it a more precise indicator of a content's potential to go viral. Its robust performance across diverse models, including XLM-Roberta, mT0, and sarvam-2b, confirms its effectiveness and applicability in practical scenarios where rapid and accurate assessment of social media content is essential.

Our analysis confirms that the ViralTweet Score (VTS) most effectively captures tweet virality, outperforming other metrics in accounting for dynamic social media interactions and biases, confirming its utility in addressing RQ1. Moreover, VTS effectively differentiates the virality of biased versus unbiased tweets, providing nuanced insights into how social biases impact virality, addressing RQ2.

As a future work, we can incorporate more complex data sources such as user demographic details and temporal engagement patterns to refine virality. Additionally, investigating the influence of external events on social media dynamics could provide deeper insights into how real-world phenomena drive online interactions. These areas not only promise to enhance the predictive power of virality metrics but also offer potential to improve strategies for content management and dissemination in digital platforms.

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

Nihar Sahoo, Niteesh Mallela, and Pushpak Bhattacharyya. With prejudice to none: A few-shot, multilingual transfer learning approach to detect social bias in low resource languages.

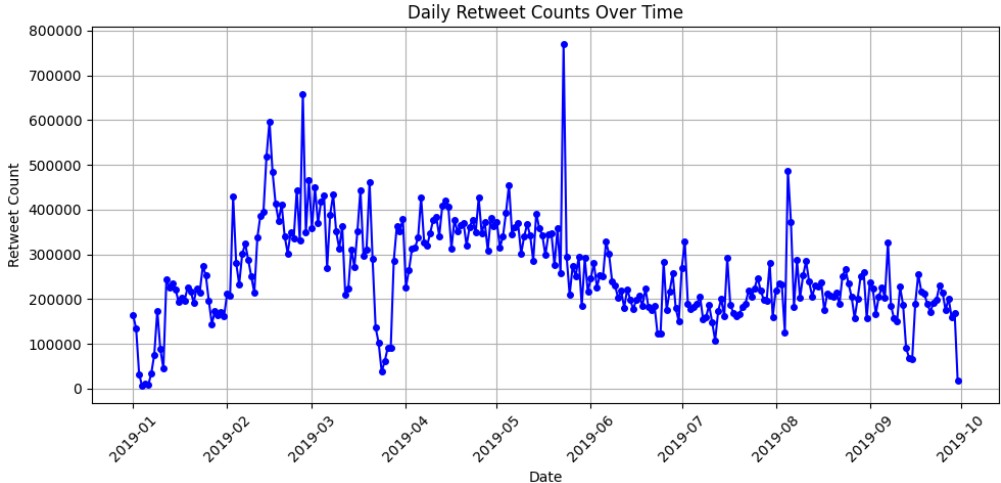

Figure 2: This figure presents a temporal analysis of Hindi tweets collected from January 1, 2019, to September 30, 2019. The daily retweet counts are aggregated in one-day bins, highlighting significant peaks on days associated with major national events. Notably, the highest peak occurs on May 23, 2019, coinciding with the Indian Election results, followed by other significant peaks on February 25, 2019, due to the Balakot airstrike, and February 14 and 15, 2019, related to the Pulwama attack. Another notable peak on August 5, 2019, corresponds to the revocation of Article 370, impacting Jammu and Kashmir's special status.

In Anna Rogers, Jordan Boyd-Graber, and Naoaki Okazaki (eds.), *Findings of the Association for Computational Linguistics: ACL 2023*, pp. 13316–13330, Toronto, Canada, July 2023. Association for Computational Linguistics. doi: 10.18653/v1/2023.findings-acl.842. URL https://aclanthology.org/2023.findings-acl.842.

Ke Wang, Mohit Bansal, and Jan-Michael Frahm. Retweet wars: Tweet popularity prediction via dynamic multimodal regression. In *2018 IEEE winter conference on applications of computer vision (WACV)*, pp. 1842–1851. IEEE, 2018.

Thomas Wolf, Lysandre Debut, Victor Sanh, Julien Chaumond, Clement Delangue, Anthony Moi, Pierric Cistac, Tim Rault, Rémi Louf, Morgan Funtowicz, Joe Davison, Sam Shleifer, Patrick von Platen, Clara Ma, Yacine Jernite, Julien Plu, Canwen Xu, Teven Le Scao, Sylvain Gugger, Mariama Drame, Quentin Lhoest, and Alexander M. Rush. Huggingface's transformers: State-of-the-art natural language processing, 2020.

Louis Wong, Ahmed Salih, Mingyao Song, and Jason Xu. Multimodal deep regression on tiktok content success. 2023.

## A  APPENDIX

| Statistic | Value |
|---|---|
| Total Tweets | 88,810 |
| Average Likes per Tweet | 3031.68 |
| Average Retweets per Tweet | 833.54 |
| Average Replies per Tweet | 194.92 |
| Average Time Series Points per Tweet | 10.259 |

Table 5: Statistics of ViralTweets Dataset. We show the total number of unique tweets in the datasets, and various statistics about the engagement metrics in the overall dataset such as average likes, retweets, and replies per tweet.

## B   DETAILS OF FINETUNING

Finetuning was performed using Huggingface library (Wolf et al., 2020) and 2 DGX A-100 cards. The Lora config for Llama and Openhathi finetunings are $r = 64, lora\_alpha = 16, dropout = 0.05, and target modules =' q_proj',' k_proj',' v_proj',' o_proj'$

## C   PROMPTS USED FOR BIAS PREDICTION

Analyze the given tweet for the presence of social biases, considering the Indian context. Social bias refers to prejudiced attitudes, stereotypes, or discriminatory behaviors that favor or disfavor certain groups based on characteristics such as religion, caste, gender, region, political beliefs, socioeconomic status, age, disability or cultural beliefs.
Given a tweet, classify whether the tweet contains social bias or not. Tweet is "Biased" if the tweet shows any form of above types of bias, otherwise it is "Unbiased".
tweet: {data_point["post"]}
label: {data_point["Bias_label"]}

## D   ANNOTATION

We engaged three annotators to label various categories in a subset of 3,000 data points. Two of the annotators are Indian females with Master's degrees, while the third is an Indian male, also with a Master's degree. Two of the annotators are aged between 26 and 30, and the third is over 40. Initially, we held two training sessions where they annotated 100 instances to familiarize themselves with the task. Once they demonstrated proficiency in the training instances, they were assigned the actual labeling task. They were provided with decent incentive for performing the task.

## E   OTHER METRICS

$$(\mathbf{VTS}_{L \times v_R})_{t+1} = likes_{t+1} \times \frac{retweets_{t+1} - retweets_t}{time_{t+1} - time_t} \tag{5}$$

$$(\mathbf{VTS}_{R \times v_L})_{t+1} = retweets_{t+1} \times \frac{likes_{t+1} - likes_t}{time_{t+1} - time_t} \tag{6}$$

$$(\mathbf{VTS}_{0.4\,R \times 0.6\,v_L})_{t+1} = 0.4 * retweets_{t+1} \times 0.6 * \frac{likes_{t+1} - likes_t}{time_{t+1} - time_t} \tag{7}$$

## F   OTHER RESULTS

Table 6: Comparative performance analysis of XLM-Roberta, mT0-large, and Sarvam-2b[12] across various virality metrics. This table showcases the mean squared error (MSE), root mean squared error (RMSE), R-squared (R2), and mean absolute error (MAE) for each model and metric. **Boldface** values indicate the best metric among the compared ones. The upward arrow ↑ indicates that a higher value corresponds to a better metric; the downward arrow ↓ indicates that a lower value corresponds to a better metric.

| Metric | XLM-Roberta | | | | mT0-large | | | | sarvam-2b | | | |
|---|---|---|---|---|---|---|---|---|---|---|---|---|
| | MSE↓ | RMSE↓ | R2↑ | MAE↓ | MSE↓ | RMSE↓ | R2↑ | MAE↓ | MSE↓ | RMSE↓ | R2↑ | MAE↓ |
| $\mathbf{VTS}_{L \times v_R}$ | 0.53 | 0.73 | 0.47 | 0.55 | 0.55 | 0.74 | 0.45 | 0.56 | 0.52 | 0.72 | 0.47 | 0.54 |
| $\mathbf{VTS}_{R \times v_L}$ | 0.53 | 0.73 | 0.46 | 0.55 | 0.56 | 0.75 | 0.43 | 0.57 | 0.51 | 0.71 | 0.48 | 0.54 |
| $\mathbf{VTS}_{F \times (v_L + v_R)}$ | 0.38 | 0.61 | 0.61 | 0.46 | 0.42 | **0.64** | **0.58** | **0.49** | 0.32 | **0.57** | **0.67** | 0.43 |
| $\mathbf{VTS}_{F \times (0.6\,v_L + 0.4\,v_R)}$ | **0.36** | **0.60** | **0.63** | **0.45** | **0.41** | **0.64** | **0.58** | **0.49** | **0.32** | **0.57** | **0.67** | **0.42** |
| $\mathbf{VTS}_{0.6\,L \times 0.4\,v_R}$ | 0.52 | 0.72 | 0.47 | 0.55 | 0.55 | 0.74 | 0.43 | 0.57 | 0.51 | 0.71 | 0.48 | 0.54 |
| $\mathbf{VTS}_{0.4\,R \times 0.6\,v_L}$ | 0.52 | 0.72 | 0.47 | 0.54 | 0.55 | 0.74 | 0.44 | 0.57 | 0.50 | 0.71 | 0.49 | 0.53 |
| **Likes** | 0.52 | 0.72 | 0.47 | 0.54 | 0.55 | 0.74 | 0.44 | 0.56 | 0.48 | 0.69 | 0.51 | 0.53 |
| **Retweets** | 0.69 | 0.83 | 0.30 | 0.64 | 0.72 | 0.85 | 0.27 | 0.66 | 0.68 | 0.82 | 0.31 | 0.64 |
| **Retweets / Followers** | 0.53 | 0.73 | 0.46 | 0.55 | 0.48 | 0.69 | 0.52 | 0.53 | 0.36 | 0.60 | 0.64 | 0.45 |
| **Likes / Followers** | 0.52 | 0.72 | 0.47 | 0.56 | 0.55 | 0.74 | 0.45 | 0.57 | 0.44 | 0.66 | 0.56 | 0.51 |

Table 7: Performance metrics (Precision, Recall, F1 Score) for different virality metrics across various Top $K\%$ thresholds. The highest values are in **bold**, and the second-highest are in *italic and underlined*. The hypothesis is that the top $K\%$ tweets based on each metric value are considered viral.

| Top $K\% \rightarrow$ | 10% | | | 15% | | | 20% | | | 25% | | |
|---|---|---|---|---|---|---|---|---|---|---|---|---|
| **Metric** | **P** | **R** | **F1** | **P** | **R** | **F1** | **P** | **R** | **F1** | **P** | **R** | **F1** |
| $\mathbf{VTS}_{L \times v_R}$ | 0.61 | 0.49 | 0.53 | 0.65 | 0.49 | 0.54 | 0.61 | 0.49 | 0.53 | 0.57 | 0.50 | 0.52 |
| $\mathbf{VTS}_{R \times v_L}$ | 0.62 | 0.50 | 0.53 | 0.65 | 0.49 | 0.54 | 0.61 | 0.50 | 0.53 | 0.57 | 0.46 | 0.49 |
| $\mathbf{VTS}_{F \times (v_L + v_R)}$ | 0.61 | 0.54 | 0.56 | 0.64 | 0.55 | 0.58 | 0.61 | 0.55 | 0.57 | 0.60 | 0.55 | 0.56 |
| $\mathbf{VTS}_{F \times (0.6\,v_L + 0.4\,v_R)}$ | 0.61 | 0.56 | 0.57 | 0.63 | 0.55 | 0.58 | 0.61 | 0.55 | 0.57 | 0.59 | 0.55 | 0.56 |
| $\mathbf{VTS}_{0.6\,L \times 0.4\,v_R}$ | 0.61 | 0.50 | 0.53 | 0.65 | 0.49 | 0.54 | 0.62 | 0.50 | 0.53 | 0.56 | 0.47 | 0.49 |
| $\mathbf{VTS}_{0.4\,R \times 0.6\,v_L}$ | 0.62 | 0.50 | 0.53 | 0.65 | 0.49 | 0.54 | 0.61 | 0.50 | 0.53 | 0.58 | 0.48 | 0.51 |
| **Likes** | 0.63 | 0.49 | 0.53 | 0.66 | 0.49 | 0.54 | 0.62 | 0.50 | 0.53 | 0.59 | 0.49 | 0.52 |
| **Retweets** | 0.61 | 0.47 | 0.51 | 0.65 | 0.47 | 0.52 | 0.61 | 0.47 | 0.51 | 0.57 | 0.47 | 0.50 |
| **Retweets / Follower** | 0.63 | 0.54 | 0.55 | 0.66 | 0.54 | 0.56 | 0.62 | 0.53 | 0.55 | 0.60 | 0.53 | 0.54 |
| **Likes / Follower** | 0.60 | 0.50 | 0.52 | 0.65 | 0.51 | 0.54 | 0.61 | 0.51 | 0.53 | 0.59 | 0.50 | 0.51 |

Table 8: Performance metrics (Precision, Recall, F1 Score) for different virality metrics across various Top $K\%$ thresholds. This is based on the experiments described in Section 6.2. The highest values are in **bold**, and the second-highest are in *italic and underlined*. The hypothesis is that the top $K\%$ tweets based on each metric value are considered viral.

| Top $K\% \rightarrow$ | 30% | | | 35% | | | 40% | | |
|---|---|---|---|---|---|---|---|---|---|
| **Metric** | **P** | **R** | **F1** | **P** | **R** | **F1** | **P** | **R** | **F1** |
| $\mathbf{VTS}_{L \times v_R}$ | 0.56 | 0.49 | 0.51 | 0.54 | 0.50 | 0.51 | 0.53 | 0.49 | 0.50 |
| $\mathbf{VTS}_{R \times v_L}$ | 0.55 | 0.49 | 0.51 | 0.53 | 0.49 | 0.50 | 0.53 | 0.49 | 0.50 |
| $\mathbf{VTS}_{F \times (v_L + v_R)}$ | 0.58 | 0.55 | 0.55 | 0.57 | 0.54 | 0.55 | 0.55 | 0.54 | 0.54 |
| $\mathbf{VTS}_{F \times (0.6\,v_L + 0.4\,v_R)}$ | 0.58 | 0.55 | 0.55 | 0.56 | 0.54 | 0.55 | 0.55 | 0.53 | 0.54 |
| $\mathbf{VTS}_{0.6\,L \times 0.4\,v_R}$ | 0.56 | 0.49 | 0.50 | 0.53 | 0.49 | 0.50 | 0.53 | 0.50 | 0.51 |
| $\mathbf{VTS}_{0.4\,R \times 0.6\,v_L}$ | 0.56 | 0.49 | 0.51 | 0.53 | 0.49 | 0.50 | 0.53 | 0.48 | 0.49 |
| **Likes** | 0.56 | 0.49 | 0.51 | 0.53 | 0.48 | 0.50 | 0.54 | 0.50 | 0.51 |
| **Retweets** | 0.54 | 0.47 | 0.49 | 0.52 | 0.47 | 0.48 | 0.51 | 0.47 | 0.48 |
| **Retweets / Follower** | 0.59 | 0.54 | 0.54 | 0.57 | 0.53 | 0.54 | 0.55 | 0.52 | 0.53 |
| **Likes / Follower** | 0.56 | 0.50 | 0.51 | 0.53 | 0.49 | 0.50 | 0.52 | 0.49 | 0.49 |

