# OpenReview forum: "MOMENTUM MEETS VIRALITY: A NOVEL METRIC FOR UNMASKING SOCIAL BIAS IN VIRAL TWEETS"
_ICLR.cc/2025/Conference — ICLR 2025 Conference Withdrawn Submission_

### Official Review · Reviewer_ZVTF · 2024-11-03

**Soundness:** 1
**Presentation:** 2
**Contribution:** 1
**Rating:** 3
**Confidence:** 4

**Summary:**

This study proposed a new metric, ViralTweet Score (VTS), to measure the virality of tweets. It was designed to capture multiple aspects of a tweet and their evolution over time. This study also proposed a dataset containing 88.8k Hindi tweets associated with VTS.

**Strengths:**

(1) This study touched on an important research topic.

(2) The manuscript is easy to follow.

**Weaknesses:**

(1) The size of the original dataset was 9.24 million. In the end, only 88.8k tweets were preserved. Would there by any selection biases during data preprocessing? The size was significantly reduced (7 million to 200 thousand) after removing tweets that did not have time-series data spanning over one day. It suggests that the remaining tweets were popular. Would this introduce biases? Next, only tweets with at least four distinct time-series data points were kept. Did this suggest that the proposed metric might not generalize well?

(2) The motivation of the study does not seem clear. In sections 3.2 and 3.3, the authors discussed content virality and virality metrics. However, it is unclear to me why the existing metrics were not good enough so that a new metric had to be proposed.

(3) The Cohen's Kappa between models and humans is low.

(4) Some text descriptions did not seem to align with the tables they referred to. "These tweets are distributed across various categories of social biases, as shown in Table 5 of Appendix A." However, Table 5 only shows the total number of unique tweets, average likes, retweets, replies, and number of time series points per tweet in the dataset. It did not relate to categories of social biases.

(5) In section 4.2, the authors discussed bias label, however, the definition of such bias was never provided. It was only until Section 4.4, the authors provided examples of the bias categories. However, why were they considered biases? Categories such as gender, religion, and race are features. It is not novel to include these additional features to improve virality prediction. There have been many efforts [1].

(6) No baseline metrics except for likes, and retweets were compared to justify the effectiveness and novelty of the proposed metric. VTS was computed based on likes and retweets. It is not surprising that it is better than either of its components. However, the performance difference is not large even comparing VTS with likes or retwets.

Referecnes:

[1] Han Y, Lappas T, Sabnis G. The importance of interactions between content characteristics and creator characteristics for studying virality in social media. Information Systems Research. 2020.

**Questions:**

Questions:

(1) "we retained only those tweets that had time-series data spanning more than one day." Can the authors clarify this?

(2) Have the authors performed any analysis to compare the characteristics of the filtered tweets versus the original dataset, or if they considered using alternative filtering criteria that might preserve a larger sample size while still meeting their data quality needs?

(3) Can the authors provide a more explicit comparison between existing metrics and the proposed VTS, highlighting the specific limitations of current approaches that VTS aims to address?

Minor:

(1) The in-text reference format seems to be incorrect. The references should be included in parenthesis.

**Details Of Ethics Concerns:**

The authors proposed to release the datasets with is composed of 88.8k tweets. Ethics review is necessary to ensure that it does not violates the copyright of Twitter.

---

### Official Review · Reviewer_GkUG · 2024-11-03

**Soundness:** 2
**Presentation:** 2
**Contribution:** 1
**Rating:** 1
**Confidence:** 4

**Summary:**

This manuscript addresses the challenge of predicting social media post virality by introducing the "ViralTweet Score (VTS)." It presents the viraltweets dataset, comprising approximately 89,000 Hindi tweets with labels gathered in 2019. The primary goal is to investigate the factors driving tweet virality in the Indian context and to explore whether virality links to social biases within the tweet. The VTS is proposed as a more effective measure of tweet virality, revealing a correlation between virality metrics and the biases present in the tweets.

**Strengths:**

- The study of what goes virality is important for understanding the current social communication landscape and societal problems.

- Non-english social media dataset is rarer and valuable.

**Weaknesses:**

- I believe that the manuscript is not a good fit for the ICLR conference because the central contribution of the paper is the ViralTweet Score (VTS) -- a hand-crafted score to measure the virality of tweets. Thus, I cannot see meaningful contribution to the field of representation learning. Although the study uses some LLMs, but the usage is rather secondary and not the main focus of the paper.

- Data collection procedure is not described in enough details. For instance, it is not clear how the initial 9.24 million tweets were selected. Was it from the streaming API or search API? How was the Hindi language tweets detected?

- The dataset includes only tweets with interactions from four specific dates—likely capturing many viral tweets, but too limited to adequately represent non-viral tweets, which could be valuable as negative or control data points.

**Questions:**

- I believe that the paper would fit much better into venues that focus on social media analysis, data mining, or computational social science, rather than NLP, ML, or representation learning venues, given its focus on the ViralTweet Score.

- To make the claim that VTS is *the* measure of assessing virality, it could be validated with other datasets and languages. Otherwise, the claim will remain very specific to this particular dataset and language.

- I think data collection procedure should be more thoroughly described and justified. For instance, the description of how the initial dataset was collected is missing nor is it clear how the Hindi language tweets were detected.

- The VTS metric is rather ad-hoc and should be justified better. There can be many different formulation to combine like and retweet counts. Why can this particular formula capture the virality best? How can we know? Can we learn this from the data?

- I think the study of bias vs. virality is too shallow and could be expanded to shed more lights on the relationship.

---

### Official Review · Reviewer_mTqQ · 2024-11-03

**Soundness:** 2
**Presentation:** 2
**Contribution:** 2
**Rating:** 5
**Confidence:** 3

**Summary:**

This paper tests existing metrics and introduce *ViralTweet Score (VTS)* to predict tweet's virality with a specific focus on social biases. VTS incorporates engagement growth over time to capture "momentum", offering a more dynamic perspective on virality in biased tweets. The authors also release the dataset and corresponding virality labels. Compared to traditional metrics, VTS more accurately predicts virality and provides insights into how social biases shape online discourse, with implications for more responsible and equitable social media analytics.

**Strengths:**

1. This paper introduces a novel metric for capturing the "momentum" of engagement, which provides a new view of virality based on the rate of engagement growth over time.
2. This paper links virality with social biases, which focuses on understanding how biased content can spread quickly on platforms and provides valuable insights into societal impacts and the role of bias in information dissemination.
3.  The paper employs both supervised and unsupervised methods to validate the effectiveness of VTS, including pairwise tweet comparisons and clustering techniques. This dual approach strengthens the evaluation of VTS, showing its effectiveness across different predictive contexts.
4. Open source the dataset, which will definitely promote relevant research in this area.

**Weaknesses:**

1. Lack of a detailed codebook for bias labels: The paper relies on social bias categories like gender, caste, and religion, but it does not provide a clear and detailed codebook or criteria for how these biases were defined and identified, especially by human annotators.
2. A little bit of over-reliance on follower count: The use of follower count as the “mass” component in VTS could bias the score towards tweets from popular accounts, which might overshadow the organic virality of tweets from less popular users.
3. Too few case studies: This paper identifies how biased tweets can become viral, but it does not delve into the practical implications of this finding, such as how VTS might inform moderation practices on social media platforms.

**Questions:**

1. Could you provide more details about the codebook or criteria used for determining social bias categories in tweets? In this paper, the authors obtain bias labels for the data through LLM models and human annotation. To validate the accuracy of these model-generated labels, you randomly select a subset of 3,000 tweets for human annotation, where three annotators with expertise in bias detection label each tweet. I think the process is fine, however, what guidelines or definitions did the annotators and models follow to identify biases?
2. Were any other engagement metrics considered in the VTS calculation (e.g., comments, views), and if not, why were they excluded?
3. For methodology, how were the initial engagement conditions selected for the pairwise comparison of tweets?
4. For the data, the paper selected Hindi tweets, but how does the focus on Indian social media impact the generalizability of the findings or there are some different findings with tweets in other language?

**Details Of Ethics Concerns:**

No ethics concerns

---

### Official Review · Reviewer_nDT3 · 2024-11-04

**Soundness:** 2
**Presentation:** 3
**Contribution:** 2
**Rating:** 3
**Confidence:** 4

**Summary:**

**Summary**
The paper explores tweet virality through a metric, ViralTweet Score (VTS), designed to better predict which tweets go viral, particularly those with social biases.

Key contributions are:

* ViralTweet Score (VTS): VTS measures virality by tracking the spread momentum of tweets over time, factoring in likes and retweets weighted by user follower count. This approach relies on the speed and extent of engagement in addition to static metrics like total likes or retweets.

* ViralTweets Dataset: The authors released a dataset of 88.8k Hindi tweets, labeled with binary bias indicators, categories of bias, and toxicity markers, to study social bias in social media virality. Labels were assigned using multiple different models including LLMs and human annotation for quality control.

* Social Bias in Virality: The study focuses on social biases inherent in viral tweets, such as biases related to gender, religion, caste, and politics, which are prevalent in Indian social media content. The authors examine whether biased tweets exhibit higher virality, using VTS as the primary measurement tool.

**Strengths:**

**Originality**
* VTS: By focusing on the rate of engagement growth (velocity) and using user follower count (mass), VTS captures virality dynamics more effectively than static metrics like total likes or retweets.

**Significance**
* Practical Relevance: Given the increasing concerns over misinformation and biased content on social media, the study’s insights are timely. VTS could help platforms and researchers identify harmful content trends more accurately.

* Focus on Social Bias: The paper addresses the socially impactful issue of bias in viral content. By measuring how bias may amplify virality, the study provides valuable insights for understanding the propagation of biased content on social media.

**Clarity**
* Writing and Organization: The paper is organized logically with minimal grammatical errors, moving from an introduction of the problem to the proposed solution, methodology, experiments, and results. Each section is well-defined, making it easy for readers to follow the research journey. I appreciate authors mentioned the filtering methodology used to filter from 9.24M tweets.

**Weaknesses:**

**Evaluation and Results**

* Lack of baselines: Although VTS outperforms some traditional metrics (likes, retweets), it lacks benchmarks from other studies and comparable metrics, even though some of them might be referenced. I advise the authors to include benchmarks from other similar studies on virality prediction [1, 2, 3] to see how VTS compares against them.

* Lack of latest related works: Recent research has moved significantly beyond purely text-based approaches, incorporating multi-modal data for virality prediction [4, 5]. The paper should compare and mention the advantages of their study relative to some of the deep learning-based approaches.

* VTS score analysis with bias clusters: The results section lacks any detailed evaluation of VTS scores in relation to bias. The study simplifies bias by categorizing tweets as either "biased" or "non-biased," without distinguishing between different types of biases (e.g., gender, religion, caste). This binary approach may obscure insights into how specific biases impact virality differently, limiting the granularity of the results. The study should provide a detailed analysis of different types of bias clusters and how they interact with virality. The authors should also discuss the limitations of VTS for different types of biases.

* Lack of generalizability :The study focuses solely on Hindi tweets, which limits the generalizability of the VTS score. The result evaluation should be conducted across multiple languages and regions to assess the value of the VTS score in broader contexts.

* Limited analysis of false positives and negatives: While the paper provides precision, recall, and F1 scores for its metrics, it lacks an in-depth analysis of false positives and false negatives in bias and virality classification. An error analysis could highlight specific areas where VTS or the bias detection model falls short, such as overestimating virality for certain topics or missing nuanced biases.

**Novelty**

* The novelty here lies in how the study tailors VTS to biased content and the Hindi social media context. The VTS itself is more of a refinement of existing temporal models.

* Other studies [6] have evaluated the spread of misinformation and biases on social media. The paper should attempt to distinguish or specify challenges related to these studies.


[1] Kwak, Haewoon, et al. "What is Twitter, a social network or a news media?." Proceedings of the 19th international conference on World wide web. 2010.

[2] Goel, Sharad, et al. "The structural virality of online diffusion." Management science 62.1 (2016): 180-196.

[3] Weng, Lilian, Filippo Menczer, and Yong-Yeol Ahn. "Virality prediction and community structure in social networks." Scientific reports 3.1 (2013): 1-6

[4] Gao, Liqun, et al. "Public opinion early warning agent model: A deep learning cascade virality prediction model based on multi-feature fusion." Frontiers in Neurorobotics 15 (2021): 674322.

[5] Zhang, Xuan, and Wei Gao. "Predicting viral rumors and vulnerable users with graph-based neural multi-task learning for infodemic surveillance." Information Processing & Management 61.1 (2024): 103520.

[6] Vosoughi, Soroush, Deb Roy, and Sinan Aral. "The spread of true and false news online." science 359.6380 (2018): 1146-1151.

**Questions:**

Some of the questions the study should try to address in future iterations of the paper.

* How to validate VTS against more complex baseline models (e.g., neural networks or multimodal models or even other temporal models) beyond traditional metrics like likes or retweets?

* How does VTS perform when applied to other types of social media content, such as images or videos? Have the study considered testing it on other platforms?

* Would a more granular approach (e.g., identifying specific types of bias) provide additional insights?

* How did the study handled hallucination and unreliability of LLM models ? Especially since most LLMs does not perform well and tend to hallucinate more on non-English content and short form content like tweets.

* How the dataset of originally 9.24M tweets was collected ? Was it truly randomly generated to preserve the distribution ?

* Does the VTS score performance differs by category of tweet ? For example, political, sports, news etc.

---

### Note · Authors · 2024-12-19

I have read and agree with the venue's withdrawal policy on behalf of myself and my co-authors.